# Proteomic Changes in Response to *Colorless nonripening* Mutation during Tomato Fruit Ripening

**DOI:** 10.3390/plants11243570

**Published:** 2022-12-17

**Authors:** Ting Zhou, Ran Li, Qinru Yu, Jingjing Wang, Jingjing Pan, Tongfei Lai

**Affiliations:** Research Centre for Plant RNA Signaling, College of Life and Environmental Science, Hangzhou Normal University, Hangzhou 310036, China

**Keywords:** tomato fruit, proteomics, *Cnr* mutant, ripening

## Abstract

*SlSPL-CNR* is a multifunctional transcription factor gene that plays important roles in regulating tomato fruit ripening. However, the molecular basis of *SlSPL-CNR* in the regulatory networks is not exactly clear. In the present study, the biochemical characteristics and expression levels of genes involved in ethylene biosynthesis in *Colorless nonripening* (*Cnr*) natural mutant were determined. The proteomic changes during the ripening stage were also uncovered by isobaric tags for relative and absolute quantitation (iTRAQ)–based quantitative proteomic analysis. Results indicated that both the lycopene content and soluble solid content (SSC) in *Cnr* fruit were lower than those in wild-type AC fruit. Meanwhile, pH, flavonoid content, and chlorophyll content were higher in *Cnr* fruit. Expressions of genes involved in ethylene biosynthesis were also downregulated or delayed in *Cnr* fruit. Furthermore, 1024 and 1234 differentially expressed proteins (DEPs) were respectively identified for the breaker and 10 days postbreaker stages. Among them, a total of 512 proteins were differentially expressed at both stages. In addition, the functions of DEPs were classified by Gene Ontology (GO) and Kyoto Encyclopedia of Genes and Genomes (KEGG) enrichment analysis. Results would lay the groundwork for wider explorations of the regulatory mechanism of *SlSPL-CNR* on tomato fruit ripening.

## 1. Introduction

Tomato (*Solanum lycopersicum*) is a widely cultivated and consumed fruit or vegetable crop. It has a high nutritional or economic value. It is often recognized as an excellent model plant for molecular genetic analysis of fleshy fruits due to the high-quality reference genome and diverse physical, physiological, and biochemical characteristics [1,2,3]. Fruit ripening is a complex and ordered process, which ultimately determines the size, shape, color, flavor, texture, nutrient content, and seed dispersal of the fruit. A combination of ripening-associated mutants and genetic engineering has permitted scientists to establish a framework for the regulation of ripening. Previous studies have shown that the ripening process of tomato fruit was coordinately controlled at the gene, transcriptional, post-transcriptional, chromatin, and epigenetic levels [4,5,6,7]. Transcription factors (TFs), phytohormones, noncoding RNAs, DNA methylation and demethylation, histone acetylation, and protein phosphorylation all play crucial roles in this complex and sophisticated regulatory network [8,9,10,11,12,13,14,15,16]. Of these, ripening-related TFs, such as MADS-box, NAC, SBP-box, HD-zip homeobox, and bHLH TFs can control fruit ripening via the involvement of phytohormone synthesis and signaling pathways (ethylene, auxin, abscisic acid, and indoleacetic acid), directly targeting ripening-related genes or interactions between proteins of the ripening regulatory network [17,18,19].

The SBP-box TFs are encoded by the *SPL* gene family that could bind to the promoter of the *SQUAMOSA* in *Antirrhinum majus* [20]. A total of 15 genes with SBP-box have been identified in tomatoes, and most of them are functionally unknown [21]. Among them, SlSPL-CNR is a multifunctional transcription factor that comprises a distinct monopartite nuclear location signal and two zinc finger motifs. It can interact with SNF1-RELATED PROTEIN KINASES (SlSnRK1) to affect cell death and tomato fruit ripening [11]. *Colorless nonripening* (*Cnr*) is a pleiotropic dominant mutation due to a spontaneous epigenetic change in the SBP-box promoter. Genetic analysis indicated that the phenotype of *Cnr* fruit was induced by hypermethylated cytosines in a recessive allele at *SlSPL-CNR*. The *Cnr* locus is mapped to the middle of the long arm of chromosome 2, and *Cnr* fruit shows a colorless, mealy pericarp with an excessive loss of cell adhesion [22,23]. So far, the molecular basis of *SlSPL-CNR* in regulating tomato fruit ripening has been rarely documented. In this study, biochemical characteristics of *Cnr* fruit were determined, and proteomic changes compared with wild-type *Solanum lycopersicum* cv. Ailsa Craig (AC) fruit during the ripening stage were acquired by isobaric tags for relative and absolute quantitation (iTRAQ)–based quantitative proteomic analysis. The differentially expressed proteins (DEGs) were quantified, identified, and functionally classified as well. The finding would lay the groundwork for additional studies of *SlSPL-CNR* function during tomato fruit ripening.

## 2. Results

### 2.1. The Biochemical Characteristics of Cnr Fruit

During fruit ripening, the biochemical characteristics of *Cnr* fruit show obvious changes in comparison with wild-type AC fruit. From the breaker stage to the 10 days postbreaker (DPB) stage, the pericarp of AC fruit gradually turned from light yellow to pink; however, *Cnr* fruit developed a colorless, mealy pericarp (Figure 1). Consistent with visual changes of physical appearance, the lycopene content and soluble solid content were lower in *Cnr* fruit than those in AC fruit along with increased days after the breaker. Contrarily, the pH, flavonoid content, and chlorophyll content were higher in *Cnr* fruit than those in AC fruit. In addition, the anthocyanin content was similar in both *Cnr* and AC fruits (Figure 2). These features indicated that *Cnr* fruit could not normally ripen, although the fruit underwent a normal growth.

### 2.2. Expression of Genes Involved in Ethylene Biosynthesis in Cnr Fruit

Ethylene plays a critical role in tomato fruit ripening. The ethylene biosynthesis is largely driven by ACC synthase (ACS) and ACC oxidase (ACO). In *Cnr* fruit, the expression levels of *SlACS1*, *SlACS3*, and *SlACS4* were significantly lower than those in AC fruit from the breaker stage to the 10 DPB stage. The expressions of *SlACS2* and *SlACS6* were significantly lower than those in AC fruit at the breaker stage and 5 DPB stages. There was no obvious expression peak for *SlACS2*, *SlACS4*, or *SlACS6* in *Cnr* fruit. In addition, the expression levels of *SlACO1*, *SlACO2*, and *SlACO3* were significantly lower than those in AC fruit at the breaker stage. Compared with AC fruit, the expression peaks of *SlACO1*, *SlACO2*, *SlACO3*, and *SlACO4* were all delayed in *Cnr* fruit (Figure 3).

### 2.3. A Global View of iTRAQ Analysis

Proteomic analysis is an important approach to offer substantial biological information on cellular events and helps to understand the accurate function of specific genes. Therefore, an iTRAQ-based quantitative proteomic analysis was performed to explore proteomic changes of *Cnr* fruit at the breaker stage and 10 DPB stage. According to the screening criteria, a total of 307,708 spectra including 68,083 annotated spectra and 58,032 unique spectra were found, and 22,840 peptides and 5214 proteins were finally identified. The peptide length and protein mass distribution, protein sequence coverage, mass error of peptide spectrum matches, and other information are shown in Appendix A and Appendix A. Through Kyoto Encyclopedia of Genes and Genomes (KEGG) pathway classification, the number of proteins involved in metabolism, carbohydrate metabolism, translation, folding, string and degradation, and amino acid metabolism was the largest (Appendix A). Through Clusters of Orthologous Groups (COG) function classification, except for the proteins with general function prediction, most of the proteins belong to the FC-O, FC-J, FC-G, FC-C, and FCE categories, and minimum proteins were related to cell motility and nuclear structure (Appendix A). Compared with AC fruit, expression levels of 510 DEPs were increased, and those of 514 DEPs were decreased in *Cnr* fruit at the breaker stage. The ratio range was 0.119 to 16.791. Meanwhile, the expression levels of 562 DEPs were increased, and those of 672 DEPs were decreased at the 10 DPB stage. The ratio range was 0.093 to 27.84 (Figure 4). The number and ratio range of DEPs at the breaker stage were both less than those at the 10 DEP stage. Furthermore, 512 proteins were differentially expressed in *Cnr* fruit at both the breaker stage and the 10 DPB stage. Among them, 69 DEPs showed a different expression trend, and others showed a similar expressing trend (166 increased and 277 decreased) in *Cnr* fruit (Figure 5 and Appendix A).

### 2.4. Function Classification of DEGs

The result patterns of Gene Ontology (GO) analysis at the breaker stage and 10 DPB stage were similar. Most DEPs belonged to the intracellular part, cytoplasm, and membrane-bounded organelle categories. The most molecular functions of DEPs were catalytic activity and binding activity. The most significant biological process enrichments were metabolic process, cellular process, and organic substance metabolic process. Furthermore, DEPs related to oxidoreductase activity and the single-organism metabolic process presented higher rich factor values for the two stages. The major difference was that more DEPs at the breaker stage participated in the small molecule metabolic process, and more DEPs at the 10 DPB stage were involved in the macromolecule metabolic process (Figure 6). For KEGG enrichments, at both stages, the number of DEPs involved in the metabolic pathways and biosynthesis of secondary metabolites was the most, and DEPs involved in glycosphingolipid biosynthesis–ganglio series possessed a higher rich factor value. Distinguishing from the 10 DPB stage, more DEPs were connected with photosynthesis at the breaker stage. For the 10 DPB stage, more DEPs involved flavone and flavonol biosynthesis, amino acid degradation, and fatty acid metabolism (Figure 7). Furthermore, DEPs belonging to the flavone and flavonol biosynthesis pathway possessed the highest rich factor values in *Cnr* fruit at the 10 DPB stage.

## 3. Discussion

In the present study, compared with AC fruit during the ripening stage, *Cnr* fruit presented lower lycopene and soluble solid contents and higher pH, flavonoid content, and chlorophyll content. These biochemical results are consistent with the nonripening phenotype of *Cnr* fruit with colorless pericarp. Meanwhile, expressions of major genes involved in ethylene biosynthesis in *Cnr* fruit were downregulated or delayed. These indicated that SlSPL-CNR is a key transcription factor and plays a critical role in tomato fruit ripening. Virus-induced *SlSPL-CNR* silencing showed inhibitory effects on fruit ripening in tomatoes [24]. Gao et al. (2019) found that the CNR CRISPR lines only showed a delayed ripening phenotype and were different from the strong nonripening phenotypes of *Cnr* natural mutants [25]. Meanwhile, SlSPL-CNR is a novel regulator of Fe-deficiency responses, and the miR157-SPL-CNR module acts upstream of bHLH101 to negatively regulated iron deficiency responses in tomatoes [26,27]. SlymiR157 could regulate *SlSPL-CNR* expression in a likely dose-dependent manner through miRNA-induced mRNA degradation and translation repression. CHROMOMETHYLASE3 was required to maintain the phenotype of *Cnr* fruit [28,29]. Therefore, the diversity and redundancy of *SPL-CNR* function in the ripening regulatory networks need to be further studied.

An iTRAQ-based quantitative proteomic analysis was used to profile DEPs in *Cnr* fruit and helped us to acquire an in-depth understanding of *SPL-CNR* on regulating fruit ripening. Through analysis of 1024 and 1234 DEPs at the breaker stage and 10 DPB stage, DEPs were located at a variety of subcellular structures, such as cytoplasm, membrane, plastid, and chloroplast. Most of the DEPs possessed catalytic or binding activity and could participate in metabolism and cellular processes. KEGG classification also proved the widespread influences of *SPL-CNR*. Thereinto, DEPs with oxidoreductase activity have higher rich factor values for both two ripening stages in GO enrichment analysis. DEPs related to glutathione metabolism were also referred to in KEGG enrichment analysis. Moreover, DEPs annotated as antioxidant enzymes (such as catalase, peroxidase, and ascorbate peroxidase) and nonenzymatic antioxidants (such as thioredoxin and glutathione) were all involved in reactive oxygen species (ROS) scavenging [30,31]. Tomato fruit ripening is an oxidative phenomenon that involves changes in the key redox homeostasis of ROS [32]. Therefore, these DEPs involved in antioxidant stress could reflect the difference in maturity between AC and *Cnr* fruits. Ethylene is a key regulatory factor at the beginning of ripening and is necessary for the process of tomato fruit ripening. Ethylene response and perception are also essential to the ripening of fruits [33]. The quantitative real-time PCR (qRT-PCR) detection results indicate that the expressions of genes involved in ethylene biosynthesis have significantly changed in *Cnr* fruit (Figure 3). However, mRNA profiling cannot capture the regulatory processes or post-transcriptional modifications that might affect the amount of active proteins. As a complement, the mount of DEPs related to ethylene production and signaling were discovered in the present study. For example, aldehyde dehydrogenase 7b, acetylornithine aminotransferase, mitogen-activated protein kinase 9, and 3-isopropylmalate dehydratase could participate in ethylene-mediated signaling pathway. In *Cnr* fruit, their expression levels were all lower than those in AC fruit. Methylthioribose-1-phosphate isomerase, calcineurin B-like calcium binding protein, glutamine cyclotransferase-like, diacylglycerol kinase, anthranilate synthase, protochlorophyllide reductase, membrane-associated progesterone receptor, mitogen-activated protein kinase 4, calcium-dependent protein kinase 1, and cell cycle control protein could respond to ethylene stimulus. Serine palmitoyltransferase, LL-diaminopimelate aminotransferase, and 1-aminocyclopropane-1-carboxylate oxidase (Solyc02g036350.2.1, SlACO6) were involved in the ethylene biosynthetic process (Appendix A). Notably, SlACO6 belongs to the iron/ascorbate-dependent oxidoreductase family and is homologous to SlACO3 [34]. Its expression level in *Cnr* fruit was higher than that in AC fruit at both the breaker and 10 DPB stages. The unexpected result could be explained that, compared with SlACO6, SlACO1 and SlACO3 maybe played a leading role in ethylene production [35,36].

In addition, the visible difference in pericarp between AC and *Cnr* fruits is mainly due to the difference in pigment content, such as chlorophylls, carotenoids, and flavonoids (Figure 2). The chlorophylls were a small group of compounds that were universally acknowledged to be indispensable photoreceptors and were intimately involved in all aspects of the primary events of photosynthesis: light harvesting, energy transfer, and conversion [37]. Carotenoids were lipophilic isoprenoid compounds that gave red color to tomato fruit and were important for plants to protect the photosynthetic apparatus against excess light [38]. The flavonoid compounds were important secondary metabolites that fulfilled a multitude of biological functions. They could regulate the development of plants by modulating auxin movement and be the effective antioxidants in photoprotection [39,40,41]. Therefore, many DEPs were involved in photosynthesis, which subsequently led to the difference in carbon fixation and TCA cycle. These results confirmed the relationships between the proteomic change and phenotypic difference between AC and *Cnr* fruits.

In addition, a total of 512 proteins were differentially expressed at both the breaker stage and 10 DPB stage in *Cnr* fruit. Most of them show similar expression trends at the two ripening stages. Generally, these DEPs were assigned into 14 functional categories based on their biological roles (Appendix A). They could participate in various biological and genetic processes, such as amino acid transport and metabolism, carbohydrate transport and metabolism, chromatin structure and dynamics, coenzyme transport and metabolism, energy production and conversion, inorganic ion transport and metabolism, and lipid transport and metabolism. Those implied the complexity of the ripening regulation network of *SlSPL-CNR*. For example, a total of 21 ribosomal proteins (including 30S, 40S, 50S, and 60S ribosomal proteins) belonging to the translation, ribosomal structure, and biogenesis category were downregulated in *Cnr* fruit. Although ribosomal proteins are known for playing an essential role in ribosome assembly and protein translation, their ribosome-independent functions have also been discovered [42]. Another, six histones, including H1 H2A, H2B, and H4, were all downregulated in *Cnr* fruit. SlH2A.Z regulates carotenoid biosynthesis and gene expression during tomato fruit ripening [43]. Various studies have suggested that fruit ripening can be driven by multiple regulators at the epigenomic level, such as DNA methylation, histone acetylation, or chromatin remodeling [4,13,16,44,45,46]. In addition, 11 DEPs belonged to cytochrome P450 enzymes’ superfamily, which were involved in the formation of membrane sterols, phytohormones and signaling molecules, biopolymers for structural support, and protection from biotic and abiotic stress in plants [47,48]. Therefore, through the careful data mining of comparative proteomics, more clues for regulating the mechanism of *SlSPL-CNR* on tomato fruit ripening could be detective. The bridges between SlSPL-CNR and the proteomic results induced by *SlSPL-CNR* silencing need to be further explored.

## 4. Materials and Methods

### 4.1. Plant Materials

AC and *Cnr* mutant were grown at 25 °C in a glasshouse with 80% humidity. The photoperiod was 16 h light/8 h dark. The flowering time was marked by the hang tag. Fruit ripening stages were recorded as days after anthesis (DPA), breaker, DPB. The breaker stage indicates that less than 10% surface of tomato fruit is pink or red.

### 4.2. Determination of Biochemical Parameters

AC fruit was harvested at the breaker, the 5 DPB and 10 DPB stages, and *Cnr* fruit was harvested at the 37, 42, and 47 DPA stages. Soluble solids content (SSC) was determined by a refractometer (LB20T, Suwei, Shanghai, China) according to product instruction [49]. The pH value was detected utilizing a pH meter (Model 3520, JENWAY, Staffordshire, UK) in 15 mL of diluted juice from 5 g crushed fruit flesh [50]. For lycopene content measurement, the fruit flesh (10 g) was ground in liquid nitrogen and repeatedly washed with methanol until the supernatant was colorless. Then, the total lycopene was extracted by 50 mL chloroform, and the absorbance of the supernatant was measured at 485 nm. Sudan I (Sangon, Shanghai, China) was as the standard substance [51]. The standard curve equation was as follows: y = 0.2764x + 0.0013, *R^2^* + 0.9954. The lycopene content was represented by μg per g fresh weight (FW).

The total flavonoid content was measured by the aluminum chloride colorimetric assay [52]. Quercetin (Sangon, Shanghai, China) was as the standard substance, and the standard curve equation was as follows: y = 0.0043x + 0.0095, R^2^ + 0.979. The fruit flesh (1 g) was ground in liquid nitrogen and suspended in 1 mL of distilled water. Then, 0.5 mL of supernatant, 1.5 mL of 95% alcohol, 0.1 mL of 10% AlCl_3_.6H_2_O, 1 mL of potassium acetate, and 2.8 mL of distilled H_2_O were mixed and incubated in the dark at 25 °C for 40 min. Subsequently, the absorbance of the supernatant was measured utilizing a SmartSpec Plus spectrophotometer (Bio-Rad, Hercules, CA, USA) at 415 nm, and the relative content of flavonoid was represented by mg per Kg FW.

The anthocyanin content was measured by the method as described by Zhang et al. (2014) [53]. Briefly, 5 g fruit flesh was ground in liquid nitrogen and diluted with methanol (containing 0.1 mol/L hydrochloric acid) to 30 mL. Let the suspension stand in the dark for 24 h and centrifuge to remove the precipitation. Subsequently, the absorbance of the supernatant was determined by a SmartSpec Plus spectrophotometer (Bio-Rad, Hercules, CA, USA) at 525 nm, and the relative content of anthocyanin was represented by absorbance per g FW. The chlorophyll content was measured according to the report of Wang et al. (2009) [54]. About 5 g of fruit flesh was ground in liquid nitrogen and diluted with 80% acetone to 30 mL. The acetone phase was collected after adequately mixing and centrifugation, and the absorbance at 645 and 663 nm was measured, respectively, for the determination of chlorophyll a and b. Total chlorophyll content was calculated using the following formula: total chlorophyll content = 8.33 × (8.02 × OD663 + 20.20 × OD645) (μg/g FW).

### 4.3. Relative Expression Analysis of Genes Involved in Ethylene Biosynthesis

Total RNAs were extracted from AC fruit at the breaker, the 5 DPB and 10 DPB stages, and from *Cnr* fruit at 37, 42, and 47 DPA, respectively, by an RNeasy Plant Mini Kit (Qiagen, Hilden, Germany). A FastQuant RT Kit (Tiangen Biotech, Beijing, China) was used for the synthesis of the first-strand cDNA. The qRT-PCR analysis was performed using 2×Ultra SYBR mixture (CW Bio, Beijing, China) in a real-time PCR amplifier (CFX96, Bio-Rad, USA). Primer pairs for the genes involved in ethylene biosynthesis were described by Lai et al. (2020) [11]. The PCR running program was 95 °C for 10 min (preheating), and 40 circles of 95 °C for 15 s, 58 °C for 15 s, and 72 °C for 20 s. The fluorescence intensity of SYBR Green in each cycle was recorded, and the threshold cycle (*Ct*) over the background was calculated by system software for each reaction. A sample was normalized using the *18S* rRNA gene, and the relative expression levels were measured using the 2^(−△Ct)^ analysis method.

### 4.4. Comparative Proteomics Analysis

AC fruit was harvested at the breaker and 10 DPB stages, and *Cnr* fruit was harvested at the 37 and 47 DPA stages. Three independent biological replicates for each sample were ground into powder in liquid nitrogen and pooled into one sample for subsequent experiments. LC-Bio of LC Science (Houston, TX, USA) provided the service of total fruit protein preparation and iTRAQ analysis. The iTRAQ labeling, SCX fractionation, and LC-ESI-MS/MS analysis based on Triple TOF 5600 were performed strictly according to the service procedure [55]. Raw data were converted into MGF files and searched using Proteome Discoverer 1.2 (PD 1.2 Thermo, Waltham, MA, USA). The Mascot search engine (Matrix Science, London, UK) was used for protein identification. The parameter setting was as follows: [Type of search: MS/MS Ion search], [Enzyme: Trypsin], [Mass Values: Monoisotopic], [Fragment Mass Tolerance: 0.1 Da], [Peptide Mass Tolerance: 0.05 Da], [Variable medications: Gln->pyro-Glu (N-term Q), Oxidation (M), iTRAQ8plex (Y)], [Instrument type: Default], [Max Missed Cleavages: 1], and [Fixed medications: Carbamidomethyl (C), iTRAQ8plex (N-term), iTRAQ8plex (K)]. For protein quantitation, a protein needs to contain at least two unique peptides. The quantitative protein ratios were weighted and normalized by the median ratio in Mascot. The proteins with *p*-value < 0.05 and ratio fold changes of >1.2 between AC and *Cnr* fruits were considered as the differentially expressed proteins. Functional annotations of the proteins were performed by the Blast2GO program against the nonredundant protein database (NR; NCBI). The COG database was used to classify and group identified proteins (http://www.geneontology.org/, accessed on 10 March 2022). In accordance with the cellular roles, each protein was assigned to 1 of the 26 functional categories (A to Z) of the COG system. The categories could provide for a detailed description of the respective COGs [56]. GO annotation and enrichment analysis of the DEPs were conducted using the WEGO software based on the Gene Ontology Consortium. The KEGG database was used to perform pathway enrichment analysis of the DEPs. The software VeNNY 2.1 was used to generate a Venn diagram, and HemI 1.0.1 was used to make a heatmap.

### 4.5. Statistical Analysis

Except for specified notification, data were pooled across three independent biological repeat experiments, and statistical analysis was performed utilizing Student’s *t*-test by the Excel software. Differences at *p* < 0.05 were considered to be significant.

## 5. Conclusions

Biochemical and proteomic changes in *Cnr* fruit during the ripening stage were determined. Compared with AC fruit, the lycopene content and SSC were lower, and the pH, flavonoid content, and chlorophyll content were higher in *Cnr* fruit. Expressions of genes involved in ethylene biosynthesis were also downregulated or delayed. Furthermore, 1024 DEPs for the breaker stage and 1234 DEPs for the 10 DPB stage were identified. Among them, a total of 512 proteins were differentially expressed in *Cnr* fruit at both the breaker stage and the 10 DPB stages. The functions of DEPs were classified by GO and KEGG enrichment analysis as well. The results would lay the groundwork for subsequent exploration of the regulatory mechanism of *SlSPL-CNR* on tomato fruit ripening.

## Figures and Tables

**Figure 1 plants-11-03570-f001:**
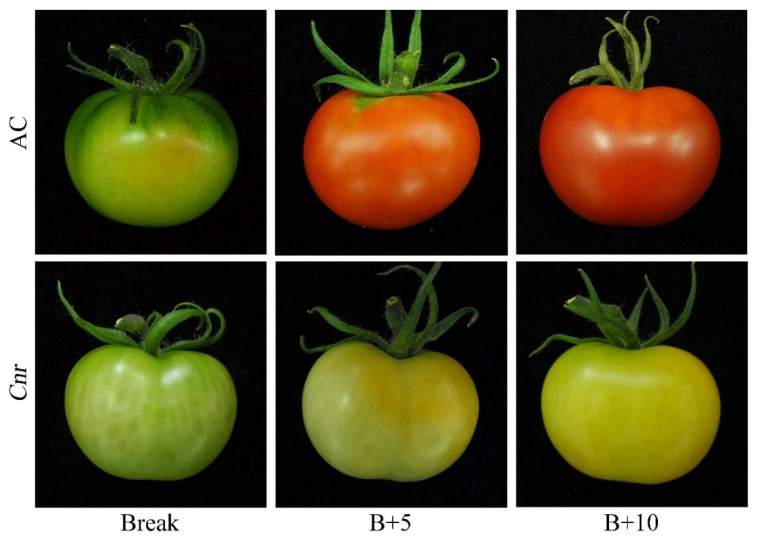
Phenotypes of wild-type AC and *Cnr* fruits at different ripening stages. B+5: 5 DPB stage; B+10: 10 DPB stage.

**Figure 2 plants-11-03570-f002:**
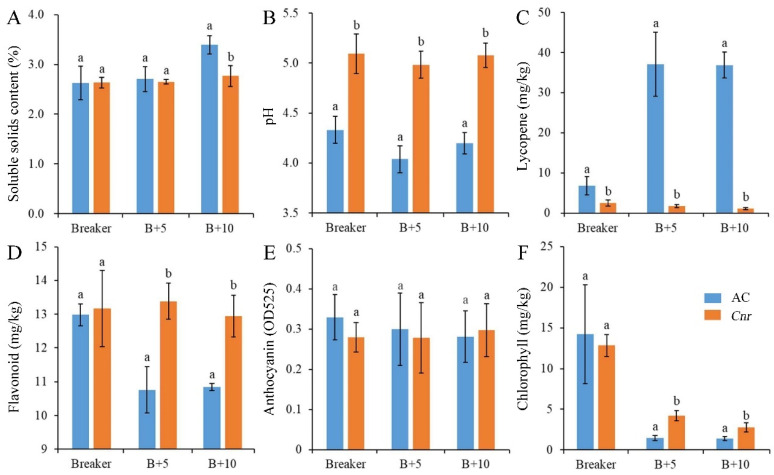
Biochemical characteristics of AC and *Cnr* fruits at different ripening stages. (**A**) Soluble solids content, (**B**) pH value, (**C**) lycopene content, (**D**) flavonoid content, (**E**) anthocyanin content, (**F**) chlorophyll content. Value represents means of three biological replicates. Bars represent standard deviation of the means. Lowercase letters a and b indicate significant difference at *p* < 0.05 based on Student’s *t*-test between AC and *Cnr* fruits at different ripening stages. B + 5: 5 DPB stage; B + 10: 10 DPB stage.

**Figure 3 plants-11-03570-f003:**
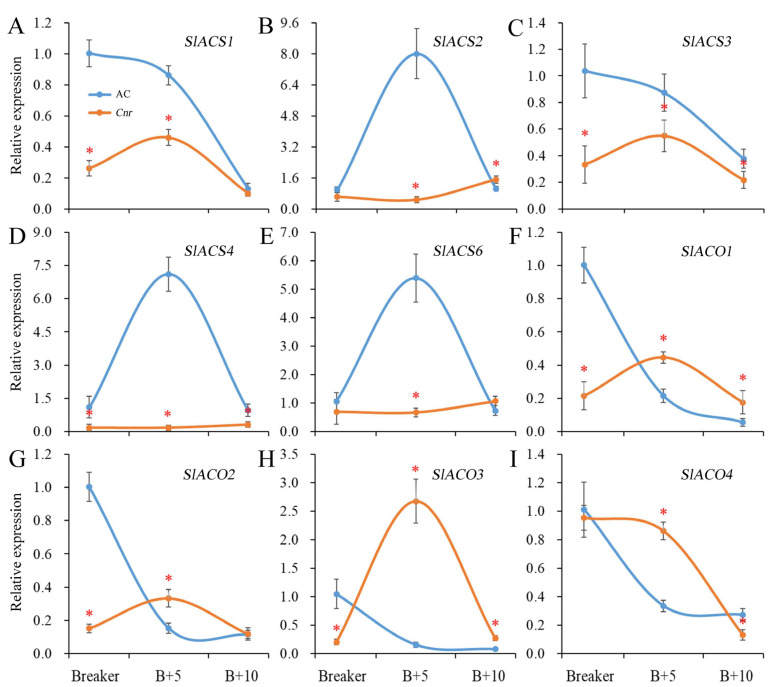
Expression of genes involved in ethylene biosynthesis in AC and *Cnr* fruits at different ripening stages. (**A**–**I**) indicate the relative expression level of *SlACS1*, *SlACS2*, *SlACS3*, *SlACS4*, *SlACS6*, *SlACO1*, *SlACO2*, *SlACO3*, and *SlACO4*, respectively. Bars represent standard deviation of the means of three biological replicates. B + 5: 5 DPB stage; B + 10: 10 DPB stage. * means significantly different (*p* < 0.05) from AC fruit based on Student’s *t*-test at each ripening stage.

**Figure 4 plants-11-03570-f004:**
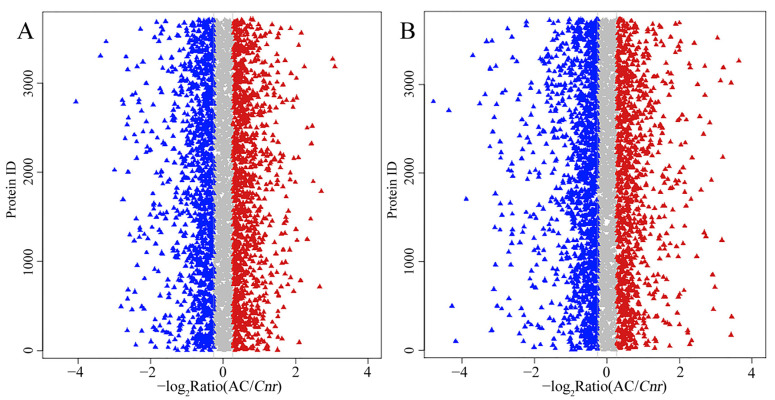
Ratio distribution of the identified proteins in AC and *Cnr* fruits at the breaker stage (**A**) and 10 DPB stage (**B**). Each triangle indicates an identified protein. Blue, gray, or red color represents decrease, no change, or increase in the expression level, respectively.

**Figure 5 plants-11-03570-f005:**
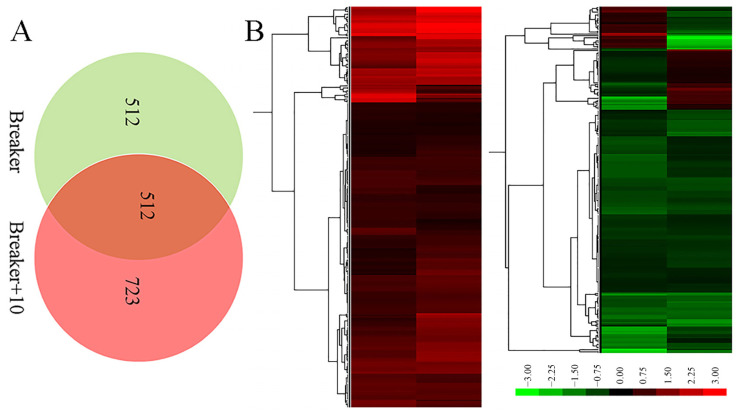
Venn diagram (**A**) and ratio heatmap (**B**) of the overlap DEPs from the breaker and 10 DPB stages between AC and *Cnr* fruits. Gradient color barcode (green to red) indicates decrease to increase in protein expression level. Each row represents a protein. Proteins with similar fold change values are clustered at the column level. The detailed original data are listed in Appendix A.

**Figure 6 plants-11-03570-f006:**
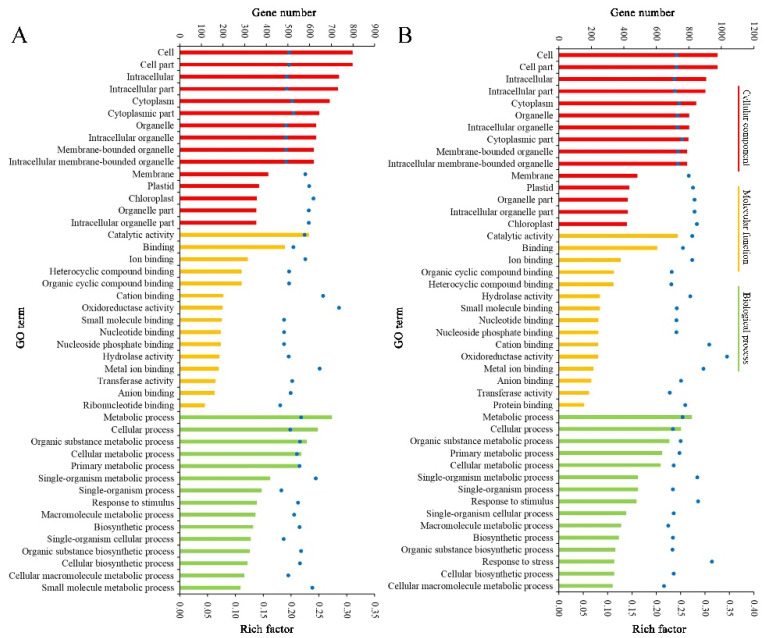
Gene Ontology (GO) enrichment of DEPs between AC and *Cnr* fruit at the breaker (**A**) and 10 DPB stages (**B**). The bar with a different color represents the gene number. The blue dot indicates the rich factor. The rich factor indicates the ratio between the number of DEPs in one GO and the number of total identified proteins in the same GO.

**Figure 7 plants-11-03570-f007:**
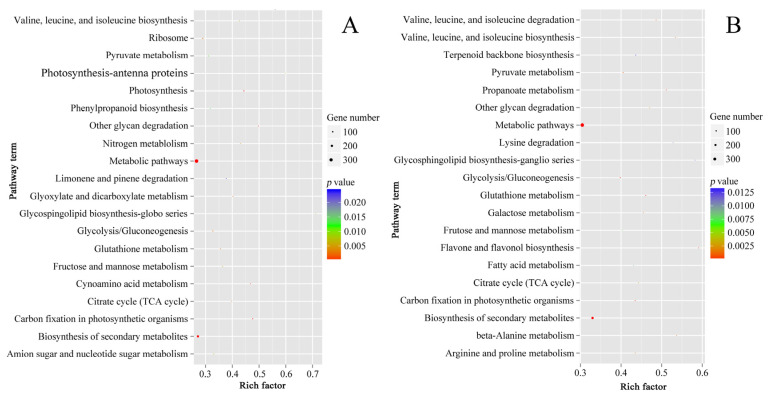
Scatter plots of the top 20 pathway enrichments of DEPs between AC and *Cnr* fruits at the breaker (**A**) and 10 DPB stages (**B**). Rich factor: The rich factor indicates the ratio between the number of DEPs in one pathway and the number of total identified proteins in the same pathway. The size of the dot represents the number of DEPs, and a larger dot indicates more DEGs.

## Data Availability

Not Applicable.

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
