# Peer review of "Proteomic Changes in Response to Colorless nonripening Mutation during Tomato Fruit Ripening"

_plants, 2022, doi:10.3390/plants11243570_

Round 1

Reviewer 1 Report

Tomato fruit has been established as a model system for studying fruit ripening, from morphology to molecular biology. In this study, the authors presented their data on biochemical characterization, gene expression, and proteomic analysis, for analyzing the Cnr mutant. The major problem is that no relationship was demonstrated among these three parts. Although the title of this manuscript says "Proteomic changes", neither biochemical analysis nor the expression of ethylene biosynthesis gene showed any relationship with the proteomic analysis. A simple mixture of three independent experiments does not make a story.

The writing itself is basically ok. However, abbreviations showed be provided in parentheses after their full names, not in the opposite way. When talking about proteins, I don't think they can be up-regulated. Only gene expression can be upregulated, unless the enzyme activity or other functions of proteins can be upregulated, not their abundance though.

Line 85, "In Cnr fruit" should be "In Cnr fruit".

Figure 2 and others should provide the statistic method in figure legends.

Figure 3 does not explain which color is for AC and which is for Cnr

Line 110, the definition of these categories should be provided.

Figure 7 has a low resolution.

Materials and Methods section should provide references for most of the methods, unless they are developed by the authors. Moreover, I don't think it an accurate way to measure lycopene, flavonoids, and anthocyanins by spectrophotometer, which is generally used a half-century ago. I could not understand how Sudan I could be used for generating a standard curve for measuring lycopene either.

Author Response

Dear reviewer,

    I would like to thank you  very much for giving us such good comments about this paper. We have earnestly revised this paper, according to the comments and suggestions made by you. The specific revisions have been highlighted in bright yellow in the revised manuscript. A detailed explanation of how we have dealt with the points raised by the reviewersws can be found in the attachment.

Best wishes!

Yours sincerely,

Tongfei Lai

On behalf of all the authors

College of Life and Environmental Science, Hangzhou Normal University, Hangzhou 310036, China

Reviewer 2 Report

This manuscript reports proteomic changes in Colorless non-ripening mutant tomato during fruit ripening. This finding contributes to our understanding postharvest physiology of tomato fruits. However, there are some issues that should be addressed.

 1. Genes involved in ethylene biosynthesis should be described correctly. For example, ACS1 needs to be rewritten to SlACS1.

 2. A sentence (line 205 to line 213) seems redundant. This sentence should be written concisely

 3. In Figure 2, Duncanʼs multiple range test was applied. Instead of this test, t-test with significance levels should be applied.

 4. In Figure 3, results of statistical analysis (t-test) with significance levels should be added.

 5. Figure 7 looks almost blank. It should be checked whether this figure is correct.

 6. English in the text can be improved.

 Overall, this manuscript has novel findings. In my conclusion, this manuscript is suitable for publication in plants if the above issues are adequately addressed.

Author Response

(The authors gave the same response as above.)

Round 2

Reviewer 1 Report

The authors have made corrections and improvements according to my comment. This reviewer does not have further concerns.